liquid biopsy

**Author for correspondence:**
Clara Chamba,
Email: clas_cha@yahoo.com

# Clinical application of circulating cell-free lymphoma DNA for fast and precise diagnosis of Burkitt lymphoma: Precision medicine for sub-Saharan Africa

Clara Chamba[1] ⬤, Sam M. Mbulaiteye[2] ⬤, Emmanuel Balandya[3] and Anna Schuh[1,4]

on behalf of the AI-REAL Consortium

[1]Department of Haematology, Muhimbili University of Health and Allied Science, Dar es Salaam, Tanzania; [2]Division of Cancer Epidemiology and Genetics, National Cancer Institute, Bethesda, MD, USA; [3]Department of Physiology, Muhimbili University of Health and Allied Sciences, Dar es Salaam, Tanzania and [4]Department of Oncology, University of Oxford, Oxford, UK

## Abstract

Burkitt lymphoma (BL) has a cure rate of around 95% when treated with chemo-immunotherapy that is standard of care in high-income countries (Minard-Colin et al., 2020, *New England Journal of Medicine* 382, 2207–2219), but currently, more than 50% of children and young adults with endemic BL (Epstein Barr virus driven BL) in sub-Saharan Africa (SSA) do not survive. Treatment for BL is largely free of charge, but there is limited access to reliable diagnostic services leading to significant delays and misdiagnoses. Innovations in histopathology such as whole slide imaging and the use of novel diagnostic approaches, in particular using circulating cell-free viral and/or lymphoma DNA (liquid biopsy), could increase access to timely and reliable diagnosis and improve outcomes in SSA.

## Impact statement

Burkitt lymphoma (BL) is a highly aggressive but curable cancer that commonly affects children. The current diagnostic pathway for BL is tissue biopsy. This is an invasive procedure that requires highly skilled personnel and specialised equipment, both of which are in short supply in sub-Saharan Africa (SSA). This results in diagnostic delays which can be as long as 2 months from the time the patient present to hospital. As a result of the highly aggressive nature of this cancer, late presentation and the diagnostic delays with tissue biopsy, the majority of children die before receiving a diagnosis. Clinicians are at times compelled to treat children based on clinical suspicion, or the less recommended method of fine needle aspiration cytology, which usually results in misdiagnosis and initiation of the wrong treatment. In our article, we highlight the need to look beyond the traditional/common methods that are used in routine clinical practice to diagnose BL and leapfrog into the advanced alternative method of 'liquid biopsy', which may be potentially faster and more precise and curb the existing diagnostic challenges. We discuss the possible implementation of this technology in SSA in view of the existing technical and infrastructural challenges that need to be considered.

## Introduction

Burkitt lymphoma (BL) is an aggressive B-cell malignancy that is more common during childhood. Historically, BL has been grouped into three clinical-epidemiological variants; endemic BL (eBL), which is found in areas with intense transmission of both Epstein Barr virus (EBV) and *Plasmodium falciparum* malaria, sporadic BL (sBL), which occurs elsewhere, and immunodeficiency BL (iBL) in immunosuppressed persons. The 2022 edition of the World Health Organization (WHO) Classification of Haematolymphoid Tumours: Lymphoid Neoplasms recommends, based on careful assessment of molecular data, that BL should be classified into EBV-positive and EBV-negative BL, based on the EBV status of the tumour, regardless of the clinical-epidemiological subgroup (Alaggio et al., 2022).

The key molecular feature of all subtypes of BL is the *c-MYC* translocation that juxtaposes the *c-MYC* gene to any of the three immunoglobulin (Ig) genes (*IGH, IGK* or *IGL*). In 80% of cases, *c-MYC* is translocated to *IGH,* and in the remaining 20%, it is translocated to one of the light chain genes, either *IGL* (15%) or *IGK* (5%). In addition to viral DNA, the presence of *c-MYC* is therefore theoretically an ideal target for the diagnosis of BL from ctDNA (God and Haque, 2010).

Endemic BL (which is reported to be ~95% EBV-positive) accounts for approximately 25–50% of childhood cancers in sub-Saharan Africa (SSA), with frequency as high as 70% in regions of SSA which are holo-endemic for malaria (Parkin et al., 2008). It is also the most common lymphoma in children worldwide (Hämmerl et al., 2019). In SSA, the incidence is highly variable from 0.5 per million in Ethiopia to 19.3 in Malawi (Hämmerl et al., 2019). In 2018, the estimated number of new cases in SSA was approximately 3,900 with the highest rate reported in males in Eastern Africa (Malawi and Uganda) and Middle Africa (Cameroon). However, due to the lack of capacity in histopathology services there remains significant uncertainty around these numbers (Hämmerl et al., 2019; Mbulaiteye and Devesa, 2022).

BL is fatal without treatment, but is routinely cured in more than 90% of cases diagnosed in high-income countries (HICs) (Minard-Colin et al., 2020), whereas cure rates are significantly lower around 30–50% for cases diagnosed in SSA (McGoldrick et al., 2019; Ozuah et al., 2020). Considering that the majority of children do not get a diagnosis, cure rates are probably lower as this number is representative of only those who received a diagnosis. The reasons for this poor outcome are poorly understood and likely to be multifactorial, but presentation with advanced-stage disease due to diagnostic delays is certainly a major contributing factor. Root causes for delays in diagnosis are thought to be a lack of awareness in the population, delays in referral to tertiary centres and lack of tertiary capacity for conventional invasive tissue diagnosis.

The latter is because the conventional diagnosis of lymphoma is by histopathology which requires an invasive excisional tissue biopsy performed by surgeons or core biopsy to be obtained by interventional radiologists followed by histopathology examination that includes formalin fixation, paraffin embedding, haematoxylin and eosin (H&E) staining and immunohistochemistry. Specialist services to achieve these at consistently high quality are limited. For example, there is less than one anatomical pathologist for every 1,000,000 people in SSA (Sayed et al., 2015; Wilson et al., 2018) leading to delays in tissue-based diagnostics to 71 days compared to the 2 days it takes to obtain results in the USA (Patel et al., 2012; Masamba et al., 2017; Fleming, 2019).

Where services are available, there is a problem with quality control. This might be due to high workload, particularly at tertiary-level care facilities, or lack of high-quality well-maintained equipment at peripheral-level care centres. Finally, automated systems, such as those routinely used to obtain reproducible immunohistochemistry are mostly lacking. These deficiencies result in delays in diagnosis, misdiagnosis and lead to incorrect treatment choices and unnecessary morbidity and mortality (Tomoka et al., 2018).

As a result, clinicians in SSA often have to make a difficult choice to wait for conventional pathology or initiate treatment on clinical suspicion, which in one series from Malawi was not confirmed by histopathology in nearly two-thirds of patients (El-Mallawany et al., 2017).

Recent innovations such as plasma genotyping of circulating cell-free DNA (cfDNA) promise to revolutionise the genetic diagnosis of solid tumours by enabling fast and non-invasive testing. For example, detection of EGFR-sensitising mutations in advanced non-small-cell lung cancer directs targeted therapy choice without the need for lung biopsy at a turn-around time of 3 days (Sacher et al., 2016). For patients with stage 2 colorectal cancer, analysis of cfDNA post-surgery has recently been shown to inform adjuvant treatment (Tie et al., 2022). Similarly, cfDNA could represents an attractive option to speed up the diagnosis of aggressive lymphomas in SSA, particularly eBL, and also for subsequent monitoring of minimal residual disease.

Here, we summarise the relevant literature of innovations in the field of cfDNA and their potential clinical applications for lymphoma diagnosis. We discuss these in the context of eBL and provide our perspective on the feasibility and future utility of this technology for eBL in SSA.

## Clinical application of cfDNA

cfDNA has potential utility in the diagnosis, prognostication and monitoring of lymphoma patients on treatment. These can be achieved via quantification of cfDNA, detection of circulating tumour DNA (ctDNA) and viral DNA (cvDNA) and quantification of these for minimal residual disease monitoring following treatment.

### Concentration of cfDNA, ctDNA mutation detection and quantification

cfDNA is composed of DNA fragments that have been released from apoptotic or necrotic cells. In healthy individuals, the normal range of cfDNA is between 1 and 10 ng/ml and is mostly derived from haematopoietic tissue. It has been established that cfDNA levels are higher and the fragment size differs in cancer patients compared to healthy controls (Marzese et al., 2013; Table 1). In individuals with cancer, these increased levels come mostly from dead tumour cells and are higher with an average of 30 ng/ml,

**Table 1.** Mean/Median cfDNA concentration comparison between cancer patients and healthy controls (measured by qPCR) (Yoon et al., 2009; Skrypkina et al., 2016; Bedin et al., 2017; Miao et al., 2019; Wu et al., 2019; Ma et al., 2020)

| Tumour type (reference) | cfDNA concentration (ng/ml) | | *p*-value |
|---|---|---|---|
| | Tumour | Healthy control[a] | |
| Hepatocellular (Ma et al., 2020) | 19.76 ± 2.68 (*n* = 84) | 5.93 ± 2.11 (*n* = 55) | <0.05 |
| Renal (Skrypkina et al., 2016) | 80.97, range 23.3–117.6 (*n* = 27) | 35.1, range 3.0–146.78 (*n* = 15) | <0.001 |
| Colorectal (Bedin et al., 2017) | 17.58, range 9.96–35.59 (*n* = 114) | 7.5, range 4.25–15.28 (*n* = 56) | <0.0001 |
| Lung (Yoon et al., 2009) | 22.6, range 3.1–730.5 (*n* = 102) | 10.4, range 1.6–89.8 (*n* = 105) | <0.0001 |
| Breast (Miao et al., 2019) | 11.97 ± 10.67 (*n* = 110) | 3.7 ± 1.24 (*n* = 95) | <0.01 |
| Lymphoma[b] (Wu et al., 2019) | 36.85 ± 46.73 (*n* = 60) | 6.67 ± 3.75 (*n* = 93) | <0.05 |

[a]Different control groups were used for the different tumour types.
[b]Lymphoma types included Hodgkin lymphoma, follicular lymphoma, natural killer T-cell lymphoma, mucosal-associated lymphoid tissue lymphoma and peripheral T-cell lymphoma.

which varies depending on the aggressiveness and size of the tumour, and whether the tumour is responding to treatment (Hohaus et al., 2013). Although identified in plasma more than 70 years ago, cfDNA has gained increasing clinical relevance only recently owing to advancements in next-generation sequencing (NGS) techniques that have enabled tumour mutation identification in cfDNA (Heitzer et al., 2019). From cfDNA, circulating tumour DNA (ctDNA) can be detected and quantified using known tumour mutation profiles and specifically for B and T-cell malignancies also the lymphoma-specific immunoglobulin gene re-arrangement. The diagnostic applications of ctDNA have been studied in a variety of cancer types (Bettegowda et al., 2014). For lymphomas with heterogeneous mutation profiles, a broad mutations cover would be required to sensitively and accurately distinguish different lymphomas (Rossi et al., 2019).

Pre-treatment levels of ctDNA show variation across the different lymphoma subtypes, with higher levels in the more aggressive subtypes, compared to the indolent subtypes. With respect to minimal residual disease, for DLBCL, a 2.5 log HGE/ml reduction in ctDNA values post two cycles of treatment is thought to be predictive of event free survival (EFS) at 24 months (Kurtz et al., 2018). Similarly, changes in ctDNA levels during therapy have also been shown to predict prognosis in patients with classical Hodgkin lymphoma (cHL; Spina et al., 2018).

For BL, historically, the gold standard method for the detection of the *c-MYC* translocation has been by fluorescence in situ hybridization (FISH). In BL, this is usually done on tissue samples. However, in addition to challenges relating to obtaining tissue samples due to the invasive nature of the procedure, it has been observed that this method can have a false negative rate (King et al., 2019) and is also not available in SSA.

Polymerase chain reaction and amplicon-based targeted NGS have been evaluated as potential alternatives to FISH for the detection of translocations. For example, for the *c-MYC* translocation, long-distance PCR (LD PCR) using specific DNA polymerases to amplify long DNA fragments, was able to detect the translocation from frozen tissue in 87% of sporadic BL cases (Basso et al., 1999). Moreover, when a sequencing-based approach for the detection of translocations using ctDNA from plasma of patients with DLBCL was assessed against conventional FISH of paired Formalin Fixed Paraffin Embedded (FFPE) samples, translocations were detected in 79% of cases (10/12 *BCL2*, 4/7 *BCL6* and 5/5 *c-MYC)* including somewhere the translocation was missed by FISH. The translocation detection rate increased to 95% in cases with high tumour burden (ctDNA > 16 pg./ml) (Kurtz et al., 2016). An important caveat is that the genomic regions affected by the translocation breakpoints have been shown to vary significantly in size between DLBCL, sporadic (mostly EBV-negative) BL and endemic (mostly EBV-positive) BL, the latter showing by far the largest region (Joos et al., 1992).

In addition to the *c-MYC* translocation, recurrent somatic mutations have been implicated in the pathogenesis of eBL, including single nucleotide variant (SNV) and insertion/deletion (indel) mutations in *c-MYC, BCL7A, BCL6, DNMT1, SNTB2, CTCF, ID3, SMARCA4, KMT2D, TCF3, TP53, DDX3X, ARID1A, CCNF* and *RHOA* (Abate et al., 2015; Grande et al., 2019; Panea et al., 2019; Burkhardt et al., 2022). For example, somatic mutations in the coding region of *c-MYC* occurred in 50% of 20 samples, *DDX3X* in 35% of 20 samples and *ID3* in 30% of 20 samples (Abate et al., 2015). More studies are needed to identify additional recurrent mutations and to understand the differences in mutation profiles between EBV-positive and EBV-negative BL.

The standard means of assessing early treatment response and monitoring for relapse in lymphoma is computed tomography (CT) and positron emission tomography (PET) scans. These are costly and in limited supply in SSA; as a result, response is often assessed based on clinical examination. Measuring ctDNA holds the promise for clinicians to recognise relapse (or primary refractory disease) before it can be detected through clinical symptoms or imaging scans – this has been evidenced in DLBCL and in follicular lymphoma through the detection of ctDNA encoding for clonally rearranged variable-diversity-joining (VDJ) region receptor genes (Kwok et al., 2016; Pott et al., 2017). Studies in cHL have shown that ctDNA may represent a biomarker for monitoring MRD in cHL through the detection of lymphoma-specific immunoglobulin gene segments (Condoluci and Rossi, 2019). More recently, the use of phased variants from ctDNA (PhasED-Seq), further improved the technical performance of ctDNA in monitoring for MRD for lymphoma (Kurtz et al., 2021).

Baseline plasma ctDNA levels have been found to correlate with clinical features and PET-CT scan features related to tumour volume/burden in DLBCL and cHL (Camus et al., 2021). In follicular lymphoma, ctDNA correlates with total metabolic tumour volume (TMTV) and adds to the prognostic value of PET-CT in identifying high-risk patients (Delfau-Larue et al., 2018). In some cancers (such as melanoma), the total ctDNA levels measured at different time points have been used to provide information on changes in tumour volume after treatment (Valpione et al., 2018). Quantification of ctDNA levels from a simple blood draw may potentially serve as an alternative measurement of tumour burden that can be used to monitor response to treatment in SSA, where PET scans are expensive and in short supply.

Studies are urgently needed to evaluate the potential role of ctDNA quantification in diagnosis and response monitoring of eBL.

### Detection and clinical application of EBV DNA

EBV infects peripheral blood mononuclear cells during the acute infection called infectious mononucleosis or glandular fever. EBV-DNA is primarily present in the cytoplasm of these cells as episomal DNA, but can also be detected in the plasma or saliva. Integration of EBV-DNA into the host genome can occur and is causally linked to the development of various EBV-associated malignancies. DNA that originates from EBV, but was integrated into host DNA can be detected in plasma of patients with EBV-related tumours.

For example, recent studies performed in patients with gastric adenocarcinomas have used cfDNA in plasma to detect EBV-DNA with results suggesting a possible role in the identification of EBV-associated gastric adenocarcinomas, predicting recurrence and response to chemotherapy (Qiu et al., 2020). Studies on EBV-positive Hodgkin's Lymphoma have demonstrated good sensitivity and specificity of plasma EBV-DNA as a non-invasive biomarker for diagnosis (Gandhi et al., 2006).

However, the use of EBV-DNA as a biomarker for diagnosis, prognostication and screening has been most extensively evaluated in nasopharyngeal carcinoma (NPC) (Lo et al., 1999; Chan et al., 2017; Tan et al., 2020).

In the landmark analysis, detection of EBV-DNA in plasma was through real-time (RT)-PCR, targeting the BamHI-W fragment and the EBNA-1 region of the EBV genome (Chan et al., 2017). The use of RT-PCR with BamHI-W as the target region, for screening the general population for NPC had a high sensitivity but a low positive predictive value (PPV) for presence of NPC when tested on

only a single occasion. There was therefore significant overlap between participants with detectable EBV-DNA in plasma and NPC, and healthy participants with detectable EBV-DNA in plasma. To improve the PPV, repeat testing with the same assay after 4 weeks, to detect persistent plasma EBV-DNA was required (Chan et al., 2017). More recently, Lam et al. demonstrated that paired-end, massively parallel sequencing-based quantification and size profiling of plasma EBV-DNA had a higher specificity and PPV compared to a single time-point RT-PCR. The study used a count-based and size-based bioinformatics approach. For a sample to be considered positive for NPC, it had to pass both the count and size-based cut-offs (Lam et al., 2018). In contrast to RT-PCR, which can only quantify EBV-DNA molecules where the amplicon region is intact, with sequencing, any EBV-DNA fragment could be analysed and counted (Lam et al., 2018). Favourable outcomes have been achieved with the employment of plasma EBV-DNA for screening and prognostication in NPC (Tan et al., 2020).

Only two studies have so far examined the role of EBV-DNA quantification for screening of eBL (Westmoreland et al., 2017; Xian et al., 2021). Both studies made use of RT-PCR to quantify EBV-DNA and found good correlation between high levels of EBV and the presence of eBL. Several confounding factors were also noted that negatively impacted on the PPV, in particular concurrent symptomatic or asymptomatic malaria parasitaemia that also led to an increase in circulating EBV-DNA. Besides, there is increasing evidence that not all eBL is EBV-driven and that spontaneous BL might be more common than previously thought in SSA (Geser et al., 1983; Alaggio et al., 2022).

Despite the scarcity of literature on the clinical application of plasma EBV-DNA in eBL, the extensive literature on plasma EBV in NPC suggests a possible clinical application for eBL. Combining specific lymphoma DNA parameters including but not limited to the *c-MYC* translocation with EBV-DNA characteristics such as sequencing-based quantification and fragment size profiling may further strengthen the possibility of early detection of EBV-driven BL through cfDNA.

## Feasibility of applying this technology in SSA

The analysis of cfDNA from a simple blood draw might be a suitable solution for SSA because the capacity building of surgical and pathological interventions poses significant challenges and will take time due to limitations in highly trained human resources (surgeons and pathologists) and infrastructure (theatres), whereas potential alternatives such as liquid biopsy -although seemingly complex and expensive- are less onerous by comparison (phlebotomy, laboratory technicians and automated sequencers).

To achieve early diagnosis of eBL in SSA using cfDNA, a diagnostic prediction model will have to be developed that employs different cfDNA parameters as shown in the graphical abstract. Before adopting cfDNA in routine diagnosis, studies need to be undertaken to technically and clinically validate this approach, to assess its health-economic impact and to evaluate its utility for eBL. To ensure long-term sustainability of cfDNA methods and affirm their clinical utility in SSA, it is paramount to conduct these studies in SSA building additional capacity in regional centres of excellence and in collaboration with national authorities.

The potential of cfDNA technology for Africa in delivering cancer diagnostics, the challenges of implementation and possible mitigating solutions have been previously highlighted (Temilola et al., 2019). However, so far, studies involving cfDNA analysis in SSA are limited. To have the necessary clinical impact, these studies will need to be conducted in SSA by researchers from SSA who can learn while doing. This approach would ensure that the technology is available to patients while it is being translated from bench to bedside and that the clinicians and technicians who will use it, develop appropriate experience before the technology is introduced for routine use.

More specifically, the use case of ctDNA in the diagnosis of eBL presents a suitable opportunity to emphasise and stress the role of genomics in cancer management, including in SSA. In SSA, despite the inclusion of genomic medicine in the curriculum for medical school training, and availability of postgraduate courses on genomic medicine, training is mostly theoretical due to lack of appropriate infrastructure to support practical training. Genomic research involving NGS, therefore, remains limited in SSA and largely restricted to collaborations at institutions of higher learning and relies on shipping samples to a well-established laboratory in a HIC, because that is where the capacity is available. We note that this practice has a counterproductive impact on local collaborators, who believe such research is esoteric, not immediately applicable to local needs, and inadvertently dampens enthusiasm to develop local capacity.

We argue that researchers in SSA should be trained to implement high-quality, reliable sample-handling processes and to conduct NGS tests locally using best practices. We also stress that south-to-south collaborations should be strengthened to build regional skills. Finally, because interaction with more established centres in HICs is critical, we recommend training for laboratory scientists through short courses to introduce specific technical skills and fellowships for more long-term mentoring with experts in centres of excellence situated in HICs. We acknowledge that achieving these goals goes beyond science and public health because it involves political aspects such as fair visa policies to access training and fair remuneration to reduce the loss of trained individuals through brain drain, as well as creation of new regional research funding streams. We believe these practical suggestions should be implemented hand-in-hand with ensuring that the necessary infrastructure, supply chains and equipment maintenance contracts are established locally to maximise the impact of the technology and the likelihood of achieving sustainability.

Considering the genetic differences that exist between EBV-positive and EBV-negative cases, diagnosis of eBL by NGS will require designing and developing bespoke targeted sequencing panels best suited to capture the genomic alterations common in eBL. Prior to implementation in routine diagnostics, the panel would need robust validation in a laboratory that is clinically accredited to ISO 15189 (quality assurance that the laboratory meets internationally recognised standards in terms of technical competence and expertise) or equivalent, based in SSA and could serve as a 'core' reference lab to future efforts. This would bridge the gap in clinical application as the clinical utility of this test is established locally.

Following a successful sequencing run, the sequencing reads generated require analysis via a specific bio-informatics pipeline. This needs advanced analytical and computational skills of highly-specialised bio-informaticians and interpretation by highly skilled individuals with both clinical knowledge and histopathological expertise. The recently established Pan-African bio-informatics network known as H3ABionet offers training programs with the aim of building capacity in bio-informatics across the African continent to enable African scientists to analyse their own data (Mulder et al., 2018). It is a first step to address the severe lack of

technical expertise in this domain across the continent. The infrastructure that has been put in place and collaborative efforts made by H3ABionet over the past few years is likely to see more institutions offering postgraduate programs in bio-informatics leading to an increase in the number of research data scientists. However, currently, most training programs are focusing on constitutional genetics and infectious disease. Our review identifies an urgent need to build capacity for analysing the cancer genomes as a logical next step.

In conclusion, cfDNA promises earlier and more precise diagnosis of eBL in SSA. The UK's National Institute of Health Research (NIHR) has funded a multicentre study in Uganda and Tanzania that evaluates this promise with initial results expected to be published in 2024 (Legason et al., 2022). In addition to a technical and clinical validation, the study addresses the need for regional multi-disciplinary diagnostic capacity building, laboratory quality control and health-economic evidence to achieve long-term sustainability and patient benefit that will ultimately catalyse multicancer early detection more generally.

**Open peer review.** To view the open peer review materials for this article, please visit http://doi.org/10.1017/pcm.2023.1.

**Acknowledgements.** The authors are thankful for the constructive comments and discussion from the members of the AI-REAL Consortium.

**Author contributions.** C.C. drafted the first version of the manuscript which was initially reviewed by members of the AI-REAL Consortium and final senior review by S.M.M., E.B. and A.S.

**Financial support.** The AI-REAL consortium is supported by a Global Health grant NIHR200133 from the National Institute for Health and Care Research (NIHR). S.M.M. is supported by the Intramural Research Program of the Division of Cancer Epidemiology and Genetics, National Cancer Institute, National Institutes of Health, Department of Health and Human Services. The contents of this publication do not necessarily reflect the views or policies of the NIHR or the NIH Department of Health and Human Services, nor does mention of trade names, commercial products or organisations imply endorsement by the US Government. The content of this manuscript is the sole responsibility of the authors.

**Competing interest.** The authors declare that they have no relevant conflict of interest.

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
