## [Reviewer Report]

*Comments to Author*: The idea of using cell-free DNA for diagnosis of Burkitt lymphoma is a very attractive one IF the technical challenges of detecting the highly variable immunoglobulin/myc translocations can be overcome. That is a big if. The authors present a well written synopsis of the rationale for this approach to diagnosis and monitoring Burkitt lymphoma. I am not sure why they are presenting this synopsis rather than the preliminary or final data since it seems they are already doing lab studies on this issue. It seems rather redundant to me but I leave this to the discretion of the journal.

Other than that, in terms of the content of the manuscript, there are several statements that the authors make that are rather controversial (or used to be controversial) largely because of scarcity of quality data from Africa. I strongly suggest they tone down these statements: (1) Burkitt lymphoma accounts for 50% - 75% of childhood cancers in Africa. Many experiences from pediatric cancer centers show 10 - 15%, with acute leukemia being the most common cancer ~30%. I appreciate that many of these are unpublished and are not population studies, but there is enough to suggest 50% - 75% is way off the mark. (2) Cure rate 40% - 50%: The actual cure rate is probably lower. I refer the authors to a recent review of published studies from Africa (Nmazuo W. Ozuah, Joseph Lubega, Carl E. Allen, Nader Kim El-Mallawany. Five Decades of Low Intensity and Low Survival: Adapting Intensified Regimens is Essential to Cure Children with Burkitt Lymphoma in sub-Saharan Africa. Blood Adv. 2020 4(16): 4007 – 4019). Also, I suggest the authors indicate that this cure rate is for about 20 - 30% of children that ever get a diagnosis and treatment; the majority of children with cancer and Burkitt lymphoma in Africa don't even get a diagnosis, which supports the rationale of their proposed diagnostic alternative.

---

## [Reviewer Report]

*Comments to Author*: Thank you for this extremely well written review. This is an important contribution to the field and, I hope, will inspire ongoing investigation in this arena. it also provides a focused look at the barriers needed to overcome and the time it will take but I agree that this is an important "leapfrog" technology. I really do not see major missing literature in this review and suggest no revisions.

---

## [Editor Report]

*Comments to Author*: Reviewer Lubega puts forth an argument to modify a couple of numbers, which I believe the authors should take under consideration and make their own decision on (although I’m unable to determine validity, the case sounds plausible). Pending this decision and slight revision if necessary, the article is ready for acceptance. The article is a very nice review, covers the topic comprehensively, in an important disease area, and I am particularly happy that it provides for an ideal exemplar for a broader discussion of the relevance of PM in underserved/under-resourced geographies.

---

## [Reviewer Report]

*Comments to Author*: The authors have reasonably addressed the issues raised in the prior version of the manuscript.

---

## [Reviewer Report]

*Comments to Author*: Useful, important review of a very important topic that is highly relevant for sub-Saharan Africa and does a great job of putting it into context of the infrastructure and human resource needs that should be paramount to make this happen.

Only a few minor revisions that I see that could strenghten the manuscript in my opinion.

1. In the Impact Statement, you note that fine needle aspiration is worse than other biopsy methods and "usually results in misdiagnosis and initiation of the wrong treatment." This may well be true but it is not comented on in the body of the article and would need references to accompany this statement. Either remove or provide evidence in the body of the manuscript.

2. On page 3, you comment that interventional radiologists are needed for core needle biopsies. This is not entirely true, especially if core needle biopsies are being done of peripheral lymph nodes. These procedures are frequently done at bedside by medical officers and clinical officers in our care.

3. In Table 1, I would relabel the second column "Cancer Patient" or something as "Tumour" is a bit misleading or at least confusing/distracting perhaps leading me to believe the measure comes directly from the tumour and not from the blood of a cancer patient.

4. On page 5, final paragraph, you being discussing ctDNA and DLBCL. There is more data in the interim years that could be included here to strengthen your argument as well. For example, https://ashpublications.org/blood/article/140/Supplement%201/1297/490222 (Herrera et al Blood 2022) and https://ashpublications.org/bloodadvances/article/6/6/1651/483728/Risk-profiling-of-patients-with-relapsed (Herrera et al Blood 2022). The literature on log change in ctDNA in many lymphoma types is rapidly evolving and promising as you say in SSA.

6. End of first paragraph on page 6, you say FISH is not available in SSA. I belive FISH is occassionally available in select centers so perhaps "routinely available" would be more approriate?

7. A figure or table demonstrating the differences in EBV measurement and testing characteristics in NPC (page 8) that could help guide similar appropriation in lymphomas would be very valuable to readers if this could be added. This is not absolutely necessary for publication but would strengthen the manuscript and broaden appeal, especially for those less familiar with the techniques who may be interested in adding to the literature in this field.

---

## [Editor Report]

*Comments to Author*: I believe this manuscript can now be accepted. The reviewer has added some additional comments which I believe can be attended by the authors post acceptance but should not delay acceptance of this article.